# Factors Predicting successful treatment outcome with novel BPaLM/BPaL regimen in individuals with drug-resistant tuberculosis: Experience From Indonesia

Arto Yuwono Soeroto[1,2], Aga Purwiga[1]*, Hendarsyah Suryadinata[1,2], Bachti Alisjahbana[1,3], Afiatin Makmun[1,4], Yana Akhmad Supriatna[1,2], Emmy Hermiyanti Pranggono[1,2], Bony Wiem Lestari[5,6]

1 Department of Internal Medicine, Faculty of Medicine Universitas Padjadjaran, Dr. Hasan Sadikin General Hospital, Bandung, West Java, Indonesia, 2 Division of Respirology and Critical Care Medicine, Faculty of Medicine Universitas Padjadjaran, Dr. Hasan Sadikin General Hospital, Bandung, West Java, Indonesia, 3 Division of Tropical and Infectious Disease, Faculty of Medicine Universitas Padjadjaran, Dr. Hasan Sadikin Bandung General Hospital, West Java, Indonesia, 4 Division of Nephrology and Hypertension, Faculty of Medicine, Universitas Padjadjaran, Dr. Hasan Sadikin Bandung General Hospital, West Java, Indonesia, 5 Department of Public Health, Faculty of Medicine Universitas Padjadjaran, Bandung, West Java, Indonesia, 6 Research Center for Care and Control of Infectious Disease, Universitas Padjadjaran, Bandung, Indonesia

☯ These authors contributed equally to this work.
* purwiga11001@mail.unpad.ac.id, aga.purwiga@gmail.com

## Abstract

### Background and aims

Drug-resistant tuberculosis (DR-TB) poses a serious challenge in Indonesia, with treatment success rates of only 50%. The BPaLM/BPaL regimen represents the latest therapeutic approach, offering shorter duration, minimal side effects, and treatment success rates of 84–90%. An operational research study conducted in Indonesia from July 2022 to March 2023 demonstrated a highly promising treatment success rate of 97.6% with the BPaLM/BPaL regimen. This study aimed to evaluate the effectiveness of the BPaLM/BPaL regimen through real-world implementation among individuals with DR-TB and identify key factors associated with successful treatment outcomes.

### Methods

This retrospective cohort study analyzed 170 DR-TB individuals treated with BPaLM/BPaL regimens at Hasan Sadikin General Hospital, a tertiary hospital in West Java, Indonesia from January 2024 to January 2025. Data were collected from the Indonesian National DR-TB registers. A successful treatment outcome was defined as completion of treatment with bacteriological response and no evidence of treatment failure. Multivariate logistic regression was used to identify factors most associated

**Data availability statement:** All relevant data are within the manuscript and its Supporting Information files.

**Funding:** This study was funded by the Academic Leadership Grant through Prof. Arto Yuwono Soeroto from the Universitas Padjadjaran (Letter Number 5317/UN6.C/PT.02/2025). The funders had no role in study design, data collection and analysis, decision to publish, or preparation of the manuscript.

**Competing interests:** The authors have declared that no competing interests exist.

with successful treatment outcomes and presented as adjusted odds ratios (aOR) with 95% confidence intervals.

## Results

The study achieved an 88.8% successful treatment rate with BPaLM/BPaL regimens. Multivariate analysis revealed treatment with no missed dose events as the most important predictor for successful treatment outcome with aOR=6.42 (95%CI 1.44–28.45, p = 0.01), followed by nutritional status improvement during treatment with aOR=3.31 (95%CI 1.06–10.37, p = 0.03). A total of 132 patients (77.6%) experienced adverse effects, predominantly gastrointestinal symptoms (93.1%) and peripheral neuropathy (30.3%). Culture conversion occurred within 2 months in 95.2% of patients, with a median time to conversion of 111 days.

## Conclusion

No missed dose events and nutritional status improvement are key predictors of successful treatment outcome. These findings support the implementation of BPaLM/BPaL regimens in Indonesia's national TB program, with emphasis on ensuring treatment adherence and nutritional status monitoring for optimal outcomes.

## Introduction

Drug-resistant tuberculosis (DR-TB) remains a major global health challenge, with Indonesia ranking second worldwide in DR-TB burden after India, accounting for 7.4% of number of incident cases globally [1]. The country reported an estimated 28,000 DR-TB cases in 2022, with treatment success rates reaching only 50%, substantially below the national target of 80% and the World Health Organization's End TB Strategy goals [2]. Traditional DR-TB treatment regimens in Indonesia have evolved from injectable-based regimens requiring 24 months to oral long-term regimens (LTR) and short-term regimens (STR) introduced in 2017. However, these approaches face significant limitations including prolonged treatment duration, severe adverse effects, poor treatment adherence, and suboptimal success rates [2,3]. Previous studies at our hospital demonstrated treatment success rates of only 50% with LTR and 72.1% with STR regimens, with major causes of failure including death and loss to follow-up [4–6].

The World Health Organization recommended in 2022 the use of Bedaquiline, Pretomanid, Linezolid, and Moxifloxacin (BPaLM/BPaL) regimen for 26–39 weeks [7,8]. Evidence supporting this regimen is based on several studies, including NIX-TB, ZENIX-Trial, and TB-PRACTECAL, which demonstrated treatment success rates of up to 90% with minimal side effects [9–11]. The BPaLM/BPaL regimen has been implemented in South Africa as a pioneer country and over 100 other nations, with Indonesia adopting it within the national TB program framework since January 2024 [1,2]. An operational research study conducted in Indonesia from July 2022 to March 2023 demonstrated a highly promising successful treatment rate of 97.6% with

the BPaL regimen among 84 drug-resistant tuberculosis individuals, significantly higher than Indonesia's current DR-TB successful treatment rate of 56% for the 2021 cohort [12].

Several factors have been identified as predictors of successful treatment in DR-TB, including younger age, male gender, adequate nutritional status, absence of previous TB treatment, low baseline acid-fast bacilli density, early sputum culture conversion, and treatment adherence [4,13–16]. However, comprehensive data on treatment outcomes and associated factors specifically for BPaLM/BPaL regimens in Indonesia remain limited. This study aimed to evaluate the effectiveness of BPaLM/BPaL regimen through real-world implementation among individuals with DR-TB Indonesia and identify key factors associated with successful treatment outcome.

## Materials and methods

### Design and setting

We conducted a retrospective cohort study among individuals aged>18 years with GeneXpert MTB/Rif confirmed rifampicin-resistant tuberculosis (RR-TB), multidrug-resistant tuberculosis (MDR-TB), and pre-extensively drug-resistant tuberculosis (pre-XDR-TB) who received BPaLM/BPaL regimens at MDR-TB Clinic of Hasan Sadikin General Hospital, a tertiary referral hospital and teaching hospital affiliated with Universitas Padjadjaran in Bandung, West Java, Indonesia. As one of the largest tertiary hospitals in Indonesia, Hasan Sadikin General Hospital serves as a regional referral center for drug-resistant tuberculosis cases from West Java and surrounding provinces. with comprehensive diagnostic facilities including rapid molecular testing (GeneXpert MTB/RIF Ultra), line probe assays, and the latest targeted next-generation sequencing (tNGS) for comprehensive drug resistance profiling which was newly implemented in 2025.

### Treatment allocation and regimen selection

All patients received treatment according to Indonesia's National DR-TB Program guidelines implemented in January 2024, which adopted the WHO recommendations for BPaLM/BPaL regimens. The choice between BPaL and BPaLM was determined by drug susceptibility testing (DST) results and baseline patient characteristics. BPaLM regimen was prescribed for patients meeting the following criteria: (1) confirmed RR-TB or MDR-TB; (2) adults and adolescents aged >14 years without pregnancy or breastfeeding status; (3) confirmed pulmonary TB or extrapulmonary TB except involving central nervous system, osteoarticular system, and miliary TB; and (4) and treatment-naive or having received prior treatment with bedaquiline, pretomanid, linezolid, or delamanid for <1 month with no documented resistance to these agents. BPaL regimen was prescribed for pre-XDR-TB patients meeting similar criteria. All treatment decisions were individualized based on the results of drug susceptibility testing and the patient's clinical characteristics.

### Inclusion and exclusion criteria

Adults (≥18 years) with GeneXpert MTB/RIF-confirmed rifampicin-resistant tuberculosis (RR-TB), multidrug-resistant tuberculosis (MDR-TB), or pre-extensively drug-resistant tuberculosis (pre-XDR-TB) who initiated treatment with BPaLM/BPaL regimen between January 1, 2024 and January 31, 2025 were included. All patients were enrolled in Indonesia's National Drug-Resistant TB Program with documented treatment outcomes available by July 31, 2025. We excluded patients with documented resistance to bedaquiline, pretomanid, or linezolid at baseline, those who discontinued treatment due to transfer-out before treatment outcome assessment and those with incomplete medical records that precluded outcome classification.

### Variables

This study evaluated eighteen factors potentially affecting treatment success in drug-resistant TB patients treated with BPaLM/BPaL regimens including: age, gender, psychiatric disorders, smoking history, body mass index (BMI), nutritional status improvement, history of previous TB treatment, comorbidities including HIV (Human Immunodeficiency

Virus), hepatitis, diabetes mellitus (DM), chronic kidney disease (CKD), hypertension, anemia, presence of cavity, baseline acid-fast bacilli (AFB) density, Xpert MTB/RIF, sputum culture conversion time, and treatment adherence (missed dose). All measurements were conducted according to the Indonesian National Drug-Resistant Tuberculosis Guideline 2024 [2].

A successful treatment outcome was defined as patients who completed treatment according to the prescribed duration with evidence of bacteriological response and no evidence of treatment failure. Unsuccessful treatment outcome include: (1) two consecutive positive cultures separated by at least 14 days, or one positive culture after confirmed culture conversion with clinical signs and symptoms of TB, or no improvement or worsening of radiological changes since baseline, (2) death from any cause during follow-up, (3) amplification of drug resistance, (4) Loss to follow-up with clinical signs or symptoms of TB (or both) when last seen, or sputum culture positive when last seen, or not sputum culture negative and with clinical signs and symptoms of TB when last seen, and (5) intolerable adverse drug reactions [2,17]. Treatment adherence was assessed based on the cumulative number of missed dose days, whether consecutive or not, during the treatment period. Patients were classified as adherent to treatment if there were no missed dose events during treatment. Nutritional status improvement was defined as an increase in BMI from treatment initiation to completion. Sputum culture conversion was defined as two consecutive negative cultures collected at least 30 days apart.

## Statistical analysis

We assessed normality of continuous variables using the Kolmogorov-Smirnov test. Normally distributed continuous variables were presented as mean ± standard deviation and compared between successful and unsuccessful treatment outcome groups using independent samples t-test. Non-normally distributed continuous variables were presented as median with interquartile ranges (IQR) and compared using Mann-Whitney U test. Categorical variables were presented as frequencies and percentages and compared using the chi-square test or Fisher's exact test (when expected cell counts were <5). We presented categorical variables as frequencies and percentages.

We performed bivariate analysis to identify potential predictors of treatment success. Variables with $p < 0.25$ in bivariate analysis were included in the multivariate logistic regression model to control for potential confounding. The multivariate model used the backward method with a significance threshold of $p < 0.05$ for retention [18]. Results were reported as adjusted odds ratios (aOR) with 95% confidence intervals (CI). Model fit was assessed using the Hosmer-Lemeshow goodness-of-fit test. All statistical analyses were performed using SPSS version 26.0, with statistical significance defined as $p < 0.05$.

## Ethics statement

This study protocol was approved by the Research Ethics Committee of Universitas Padjadjaran (approval number 74/UN6.KEP/EC/2024). All patients included in this study were enrolled in Indonesia's National Drug-Resistant Tuberculosis Program, where BPaLM/BPaL treatment is provided as standard of care following national treatment guidelines. As this study involved routine programmatic data collection from patients receiving standard treatment within the national TB control program framework, and given the retrospective nature of the analysis using de-identified routine program data, the ethics committee waived the requirement for individual informed consent. Patient anonymity was preserved through data de-identification, and strict confidentiality measures were implemented throughout the data collection and analysis process.

## BPaLM/BPaL program implementation at Hasan Sadikin General Hospital

Hasan Sadikin General Hospital implemented the BPaLM/BPaL regimen within a structured DR-TB management program following its adoption into Indonesia's national tuberculosis program in 2024. The program operates under the following

framework: Treatment decisions are made by a multidisciplinary Clinical Expert Team comprising specialists from pulmonology, internal medicine, clinical pathology, radiology, psychiatry, and clinical pharmacologist, with consultation from other relevant specialties as indicated. All treatment regimens are individualized based on drug susceptibility testing results and patient clinical conditions

Upon confirmation of RR/MDR-TB diagnosis, patients receive treatment initiation within 7 days. The program follows an ambulatory care model where patients receive outpatient-based treatment with routine monthly monitoring visits, which can be intensified to more frequent intervals when clinically indicated (e.g., adverse events, adherence concerns, or clinical deterioration). The program implements systematic monitoring throughout the treatment continuum, including initial comprehensive baseline assessment, monthly clinical and laboratory evaluations (complete blood count, liver and renal function tests) with adherence monitoring, quarterly radiological assessments, and post-treatment follow-up. Patients receive comprehensive support including monthly financial assistance through Indonesia's National TB Program, treatment adherence counseling, nutritional support, and access to psychiatric consultation when indicated.

## Results

### Baseline characteristics

A total of 170 drug-resistant tuberculosis patients who received BPaLM/BPaL regimens were included in this study, as depicted in Table 1. All patients completed their treatment by the end of July 2025, with documented treatment outcomes available for analysis (Fig 1). Patients had a median age of 38.0 years, with the majority (69.4%) being ≤45 years old and predominantly male (60.0%). More than half of patients (52.9%) were underweight at treatment initiation, with a median BMI of 18.4 kg/m². Half of the patients (50.6%) had a history of previous TB treatment. Diabetes mellitus was the most common comorbidity, affecting 21.2% of patients, while other comorbidities, including HIV, hepatitis, and chronic kidney disease, had very low prevalence (<2%). Early culture conversion occurred in most patients, with 95.2% achieving conversion within the first two months of treatment. Treatment adherence was high, with 94.7% of patients having no missed doses during treatment. The majority of patients (77.1%) received the BPaLM regimen. Overall treatment success was achieved in 151 patients (88.8%). Among the 19 patients who experienced unsuccessful treatment outcome, including: death from any cause (57.8%), loss to follow-up (15.7%), and amplification of drug resistance (26.3%). A total of 132 patients (77.6%) experienced adverse effects, predominantly gastrointestinal symptoms (93.1%) and peripheral neuropathy (30.3%).

### Multivariate analysis

Bivariate analysis identified several variables with potential associations with treatment success (p < 0.25), including younger age (≤45 years), male gender, improvement in nutritional status, and no missed dose events during treatment, as presented in Table 2. In the final multivariate model, two factors emerged as independent predictors of treatment success. Treatment adherence was the strongest predictor, with patients who had no missed doses showing significantly higher success rates compared to those with missed doses (aOR = 6.42, 95% CI 1.44–28.5, p = 0.01). BMI improvement during treatment was the second significant predictor (aOR=3.31, 95%CI 1.06–10.37, p = 0.03).

## Discussion

### Factors associated with treatment success

This retrospective cohort study represents the first comprehensive real-world evaluation of BPaLM/BPaL regimen implementation in Indonesia following its adoption into the national tuberculosis program in January 2024. The timing of this research is particularly significant given Indonesia's position as the second-highest burden country for drug-resistant tuberculosis globally. The study achieved an overall treatment success rate of 88.8% demonstrating remarkable

**Table 1. Characteristics of study subjects.**

| Variables | Number (%)[a]<br>N = 170 |
|---|---|
| Age (years) | |
| ≤45 | 118 (69,4) |
| >45 | 52 (30,6) |
| Gender | |
| Male | 102 (60,0) |
| Female | 68 (40,0) |
| Psychiatric Disorders | |
| None | 168 (98,8) |
| Present | 2 (1,2) |
| Smoking History | |
| Never | 86 (50,6) |
| Former/Current | 84 (49,4) |
| Body Mass Index (BMI) | |
| Median (Q1–Q3) | 18,42 (15,60–21,70) |
| Underweight | 90 (52,9) |
| Normal | 51 (30,0) |
| Overweight | 13 (7,6) |
| Obese | 16 (9,4) |
| History of Previous TB Treatment | |
| None | 84 (49,4) |
| Present | 86 (50,6) |
| Comorbidities (Present) | |
| HIV | 1 (0,6) |
| Hepatitis | 1 (0,6) |
| Diabetes Mellitus | 36 (21,2) |
| Chronic Kidney Disease | 3 (1,8) |
| Hypertension | 11 (6,4) |
| Hemoglobin (g/dl) | |
| Median (Q1-Q3) | 12,8 (11,5–14,1) |
| Anemia | 100 (58,8) |
| No Anemia | 70 (41,2) |
| Cavity | |
| None | 105 (61,8) |
| Present | 65 (38,2) |
| Baseline AFB Smear | |
| Negative | 103 (60,6) |
| Scanty | 6 (2,9) |
| +1 | 37 (21,8) |
| +2 | 19 (11,2) |
| +3 | 6 (3,5) |
| Baseline Culture | |
| Negative | 29 (17,1) |
| Positive | 141 (82,9) |
| *Xpert MTB/RIF (n=115)* | |
| *Very Low* | 10 (9,5) |

*(Continued)*

Table 1. (Continued)

| Variables | Number (%)[a] N = 170 |
|---|---|
| *Low* | 30 (28,6) |
| *Medium* | 22 (21,0) |
| *High* | 43 (41,0) |
| Time to Culture Conversion (n=162) | |
| Month 0 | 27 (16,7) |
| Month 1 | 88 (54,3) |
| Month 2 | 45 (27,8) |
| Month 3 | 2 (1,2) |
| Month 4 | - |
| *Missed Dose* (days) | |
| Mean±SD | 1,64 ± 10,0 |
| Present | 161 (94,7) |
| Absent | 9 (5,3) |
| Interval from Clinical Team Decision to Treatment Initiation (Days) | |
| Median (Q1–Q3) | 5,0 (5,0 – 6,25) |
| Two Consecutive Culture Conversion (days) | |
| Median (Q1–Q3) | 111,0 (105,0–136,0) |
| Regimens | |
| BPaL | 39 (22,9) |
| BPaLM | 131 (77,1) |
| Outcome | |
| Unsuccessful | 19 (11,2) |
| Successful | 151 (88,8) |

[a] Numerical data presented as mean±SD, median (Q1–Q3)

consistency with NIX-TB, ZENIX-Trial, and TB-PRACTECAL studies and representing a substantial improvement over Indonesia's historical outcomes with longer-term regimens (50%) and short-term regimens (72.1%) [4,5]. The adoption of BPaLM/BPaL represents a paradigmatic shift from prolonged, injectable-based treatments to shorter, all-oral regimens with demonstrated efficacy rates of 84–90% in clinical trials [9–11].

Several previous studies have shown factors that influence DR-TB treatment outcomes, including: age<45 years, male gender, nutritional status, previous TB treatment history, AFB density, treatment adherence, sputum conversion time, and several comorbidities (HIV, hepatitis, diabetes mellitus, and chronic kidney disease) [4,13–16]. These various factors interact with each other to influence treatment outcomes through mechanisms of immune system suppression, impaired drug absorption, increased drug adverse events, higher disease burden, and poor treatment adherence [19–26]. While numerous variables were analyzed in this study, several deserve particular attention beyond the statistically significant predictors.

Multivariate analysis identified two independent predictors of treatment success, both representing modifiable factors amenable to programmatic intervention. Treatment adherence emerged as the strongest predictor, with patients experiencing no missed doses demonstrating significantly higher success rates compared to those with missed doses (aOR=6.4). The study population exhibited exceptional adherence, with 94.7% of patients having no missed doses during the critical treatment period, likely contributing to the overall high success rate. The complexity of DR-TB treatment

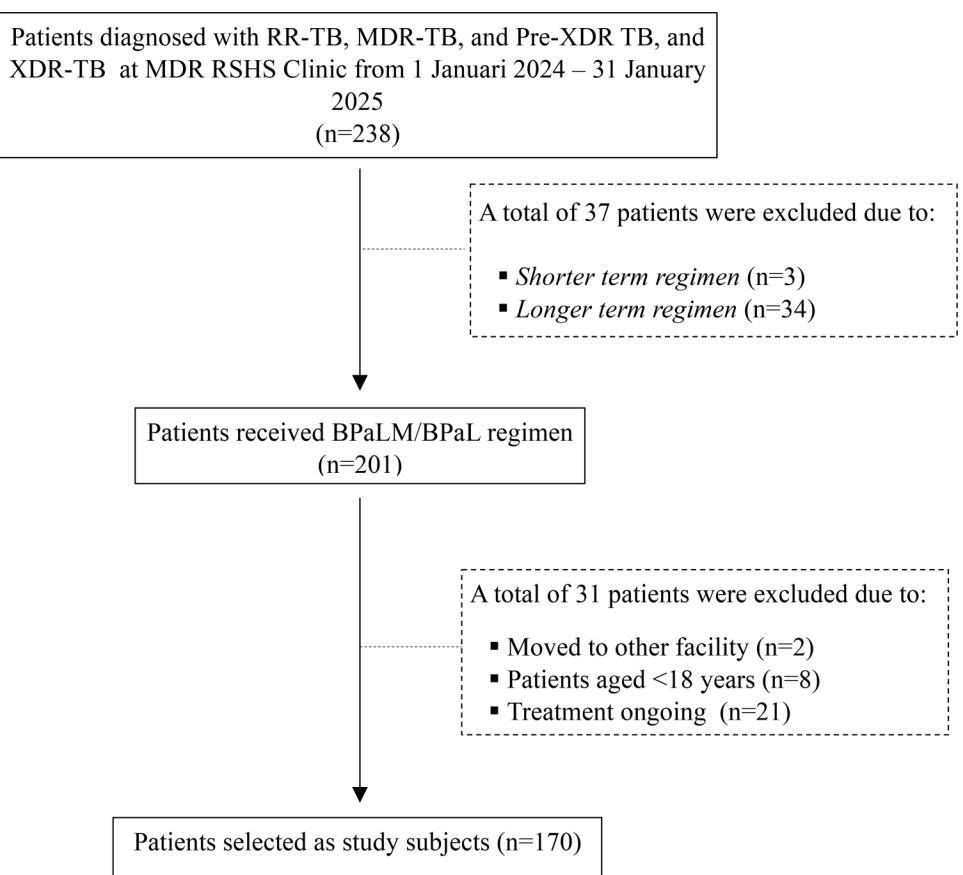

**Fig 1. Study enrollment flowchart.** Selection of 170 drug-resistant tuberculosis patients treated with BPaLM/BPaL regimens at Hasan Sadikin General Hospital, Bandung, Indonesia, 2024–2025..

involves non-medical aspects that contribute to treatment success [16]. During treatment, patients often experience treatment fatigue, develop non-adherence, and discontinue medication due to adverse effects, ultimately leading to treatment failure [8]. Non-adherence causes treatment failure, severe disease, prolonged hospitalization, increased treatment costs, and increased risk of drug resistance, making it crucial to recognize factors associated with treatment adherence [27]. Several programmatic factors may explain the high adherence observed in our study. Patients at our MDR-TB clinic received comprehensive support, including monthly financial support (IDR 200,000, ≅ 12.18 USD), addressing economic barriers that commonly contribute to treatment adherence [16]. Our results reinforce treatment adherence as the most critical biopsychosocial determinant of treatment success.

The second significant predictor was nutritional status improvement during treatment (aOR=3.3). Malnutrition is a risk factor for low cure rates and high mortality among DR-TB patients [6]. Death is significantly higher in patients with a BMI less than 18.5 kg/m² (aOR=2.73, 95% CI 1.01–7.39; P<0.048) [28]. Malnutrition causes dysfunction of liver and kidney function that interferes with drug metabolism and excretion, potentially causing drug toxicity or sub-therapeutic drug levels. Malnutrition reduces the effectiveness of anti-tuberculosis treatment and weakens patients' immune systems by disrupting the function of T cells, macrophages, and natural killer cells [22,23,29]. This study also indicated that there was no significant difference found in BMI at baseline between the successful and unsuccessful groups. BMI improvement during

**Table 2. Multivariate analysis of factors most associated with successful treatment outcome.**

| Variables | Initial Model | | Final Model | |
|---|---|---|---|---|
| | Crude OR (95%CI) | p-value | Adjusted OR (95%CI) | p-value |
| Age (years) | | | | |
| ≤45 | Reference | 0.09[a] | Reference | 0.05 |
| >45 | 0.44 (0.16–1.16) | | 2.83 (0.99–8.11) | |
| Gender | | | | |
| Male | Reference | 0.09[a] | Reference | 0.19 |
| Female | 0.44 (0.16–1.16) | | 2.01 (0.70–5.78) | |
| Psychiatric Disorders | | | | |
| None | Reference | 1.00[b] | | |
| Present | 0.88 (0.84–0.93) | | | |
| Smoking History | | | | |
| Never | 0.99 (0.89–1.10) | 0.85[b] | | |
| Former/Current | Reference | | | |
| Body Mass Index (BMI) | - | 0.09[a] | | |
| Median (Q1–Q3) | | | | |
| Underweight | | | | |
| Normal | | | | |
| Overweight | | | | |
| Obese | | | | |
| Nutritional Status Improvement | | | | |
| None | Reference | 0.03[b]* | Reference | 0.03* |
| Present | 3.20 (1.08–9.43) | | 3.31 (1.06-10.37) | |
| History of Previous TB Treatment | | | | |
| None | Reference | 0.20 | | |
| Present | 0.53 (0.19–1.42) | | | |
| Comorbidities (Present) | | | | |
| HIV | 0.88 (0.84–0.93) | 1.00[b] | | |
| Hepatitis | 0.88 (0.84–0.93) | 1.00[b] | | |
| Diabetes Mellitus | 1.38 (0.46–4.16) | 0.56[a] | | |
| CKD | 0.88 (0.83–0.93) | 1.00[b] | | |
| Hypertension | 0.87 (0.82–0.93) | 0.10[b] | | |
| Anemia | | | | |
| None | Reference | 0.93[a] | | |
| Present | 1.05 (0.39–2.77) | | | |
| Cavity | | 0.89[a] | | |
| None | Reference | | | |
| Present | 0.94 (0.34–2.56) | | | |
| Baseline AFB Smear | - | 0.90[b] | | |
| Negatif | | | | |
| *Scanty* | | | | |
| +1 | | | | |
| +2 | | | | |
| +3 | | | | |
| Xpert MTB/RIF (n=105) | - | 0.48[b] | | |
| *Very Low* | | | | |

*(Continued)*

**Table 2.** (Continued)

| Variables | Initial Model | | Final Model | |
|---|---|---|---|---|
| | Crude OR (95%CI) | p-value | Adjusted OR (95%CI) | p-value |
| *Low* | | | | |
| *Medium* | | | | |
| *High* | | | | |
| Sputum Conversion | | | | |
| ≤2 Months | 0.93 (0.90–0.97) | 1.00[b] | | |
| >2 Months | Reference | | | |
| *Missed Dose* | | | | |
| None | 7.81 (1.88 – 33.33) | 0.01[b*] | 6.42 (1.44 – 28.45) | 0.01* |
| Present | Reference | | Reference | |

Outcome: Successful treatment

*)Statistically significant

BMI: Body Mass Index; HIV: human immunodeficiency virus; CKD: chronic kidney disease; AFB: acid-fast bacilli

represents a novel finding that extends beyond simple baseline nutritional status. A routine nutritional status assessment during treatment follow-up may be more clinically relevant than static baseline measurements [6,30].

Our study found that 162 (95.2%) patients experienced culture conversion within the first two months. These results are consistent with several studies that demonstrate the importance of early culture conversion timing. A prospective cohort study by Kurbatova et al. among 1,712 DR-TB patients found that culture conversion within the first two months had higher treatment success rates with aOR=4.12 (95% CI 2.25–7.54).[31] Early culture emerged as a strong predictor of overall success, confirming its value as an early marker of treatment effectiveness and supporting current monitoring protocols.

## Operational challenges and programmatic implementation

The successful implementation of BPaLM/BPaL regimens in our setting was supported by key programmatic elements relevant to national scale-up. In Indonesia, DR-TB medications are centrally procured through the Ministry of Health and distributed via Provincial Health Offices to designated referral facilities, ensuring consistent drug availability at our center. A major challenge in resource-limited settings is access to treatment centers, as the number of DR-TB referral facilities remains limited, and patients must travel considerable distances for monthly monitoring. To address this, our hospital collaborates with a non-governmental organization funded by the Stop TB Partnership Indonesia (STPI) to provide monthly financial aid of IDR 200,000 (approximately USD 12.18) per patient for transportation and nutritional support. This support mechanism appeared instrumental in achieving high adherence rates, with 88.2% of patients having no missed dose events.

The high treatment success rate observed in our study (88.8%) may be attributed to several programmatic factors implemented at our center. The median time from diagnosis to treatment initiation was 5.0 days, with 80.6% of patients beginning treatment within 7 days. This early treatment may minimize disease progression during the vulnerable pre-treatment period. The ambulatory care model, with monthly comprehensive monitoring, may have contributed to the early detection of adverse events and timely intervention for adherence challenges. Financial support provided through the national program likely addressed economic barriers that commonly contribute to treatment default in resource-limited settings. However, the independent contribution of these programmatic elements to treatment success requires further investigation through comparative studies. Future research should examine: (1) the relationship between early case detection, rapid treatment initiation, and treatment success rates, (2) the cost-effectiveness of various patient support mechanisms in improving

adherence, (3) the optimal frequency and timing of monitoring visits to maintain treatment quality while minimizing patient burden, and (4) the applicability and replicability of this programmatic model in diverse resource-limited settings.

Our findings reinforce that effective drug-resistant TB treatment requires not only access to quality medications but also comprehensive patient support systems. Previous studies have consistently identified treatment adherence as a critical determinant of treatment success. The combination of reliable drug supply and financial support for patients in our program demonstrates that well-designed regimens, when implemented with adequate programmatic support, can achieve high treatment success rates even in resource-limited settings. These programmatic elements should be considered essential components when planning the national scale-up of BPaLM/BPaL regimens, particularly in settings with similar economic and geographical challenges.

## Study limitations

This study has several important limitations. First, our definition of treatment success was based on end-of-treatment outcomes rather than long-term follow-up, which may not capture treatment sustainability. Long-term assessment of DR-TB treatment outcomes represents a more objective parameter for evaluating treatment success compared to definitions that only assess results on the final day of treatment [32]. In 2021, the World Health Organization proposed the term "sustained treatment success" for use in operational research. In the context of DR-TB, this term refers to individuals with DR-TB who remain TB-free after 6 months and 12 months of post-treatment follow-up monitoring [33]. This study found that 99 (65.5%) of the 151 patients who achieved treatment success underwent repeat culture testing at six months post-treatment, with negative culture results.

Second, the relatively small number of unsuccessful treatment events (n = 19) limits the precision of effect size estimates, as evidenced by wide confidence intervals in the multivariate analysis (e.g., aOR for adherence: 1.44–28.5). While this reduces statistical precision for individual predictors, we chose to include multiple predictors based on established literature documenting the multifactorial nature of DR-TB treatment outcomes. A systematic review and meta-analysis conducted by Johnston et al. has consistently demonstrated that successful DR-TB treatment is influenced by numerous interrelated factors across demographic, nutritional, treatment-related, clinical, comorbidity, and behavioral domains [34]. From a clinical and programmatic perspective, oversimplifying the model to one or two predictors for statistical parsimony would not adequately capture the complexity of the real-world determinants of treatment success that clinicians and public health practitioners must consider when identifying high-risk patients, designing interventions, and allocating programmatic resources.

Third, potential selection bias exists due to the retrospective design and non-randomized treatment allocation. Regimen selection was based on drug susceptibility testing and clinical judgment, which may have introduced systematic differences between groups. Additionally, treatment at a tertiary referral center may have selected for specific patient populations, limiting generalizability.

Fourth, the low prevalence of important comorbidities such as HIV (0.6%), hepatitis (0.6%), and chronic kidney disease (1.8%) prevented adequate assessment of their impact on treatment outcomes, limiting generalizability to populations with higher comorbidity burdens. Finally, the relatively short implementation period (January 2024 to June 2025) may not capture seasonal variations or longer-term programmatic challenges that could affect treatment outcomes in routine practice. Despite these limitations, the study provides valuable real-world evidence of BPaLM/BPaL's effectiveness in the Indonesian healthcare context following its programmatic rollout in January 2024.

## Conclusion

BPaLM/BPaL regimen demonstrates high treatment success rates in Indonesian DR-TB patients. No missed-dose events and improvement in nutritional status were the most important predictors of successful treatment outcome. These findings

emphasize the importance of comprehensive patient support systems to ensure treatment adherence, nutritional support, and monitoring for optimal outcomes.

## Supporting information

**S1 Table. Treatment-related side effects.**
(DOCX)

## Author contributions

**Conceptualization:** Aga Purwiga, Arto Yuwono Soeroto, Bony Wiem Lestari.

**Formal analysis:** Aga Purwiga, Bony Wiem Lestari.

**Writing – original draft:** Aga Purwiga, Arto Yuwono Soeroto, Bony Wiem Lestari.

**Writing – review & editing:** Hendarsyah Suryadinata, Bachti Alisjahbana, Afiatin Makmun, Yana Akhmad Supriatna, Emmy Hermiyanti Pranggono,.

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
