## [Decision Letter · Decision Letter 0]

11 Dec 2025

Dear Dr. Purwiga,

Thank you for submitting your manuscript to PLOS ONE. After careful consideration, we feel that it has merit but does not fully meet PLOS ONE’s publication criteria as it currently stands. Therefore, we invite you to submit a revised version of the manuscript that addresses the points raised during the review process.

**ACADEMIC EDITOR:** Thank you for your submission. As noted by the reviewers, it has significant merit and relevance. In addition to the points raised by the reviewers, please also consider whether the section on "Clinical implementation and programmatic success factors" may instead be better placed in the background section. At current, it reads as your reflections on why there is such high success with BPAL / BPALM in your setting. But what is the justification for your belief that these are the four reasons? To avoid the speculative component, it would be stronger to describe HOW the new regimens were implemented in the background / context. And include some of the reflections as a much shorter, more circumspect set of propositions for further investigation in the discussion. This would also create space for the reviewers' requests for more information - see their comments.Thank you for your submission. As noted by the reviewers, it has significant merit and relevance. In addition to the points raised by the reviewers, please also consider whether the section on "Clinical implementation and programmatic success factors" may instead be better placed in the background section. At current, it reads as your reflections on why there is such high success with BPAL / BPALM in your setting. But what is the justification for your belief that these are the four reasons? To avoid the speculative component, it would be stronger to describe HOW the new regimens were implemented in the background / context. And include some of the reflections as a much shorter, more circumspect set of propositions for further investigation in the discussion. This would also create space for the reviewers' requests for more information - see their comments.

We look forward to receiving your revised manuscript.

Kind regards,

Graeme Hoddinott, Ph.D

Academic Editor

PLOS One

Journal Requirements:

https://journals.plos.org/plosone/s/file?id=ba62/PLOSOne_formatting_sample_title_authors_affiliations.pdf ..

“This study was funded by the Academic Leadership Grant through Prof. Arto Yuwono Soeroto from the Universitas Padjadjaran (Letter Number 5317/UN6.C/PT.02/2025)”

3. Please be informed that funding information should not appear in the Acknowledgments section or other areas of your manuscript. We will only publish funding information present in the Funding Statement section of the online submission form. Please remove any funding-related text from the manuscript.

4. Please upload a copy of Figure 1, to which you refer in your text on page 11. If the figure is no longer to be included as part of the submission please remove all reference to it within the text.

5. Please include a copy of Table 1 & 2 which you refer to in your text on page 11 & 12.

Reviewers' comments:

Reviewer's Responses to Questions

**Comments to the Author**

1. Is the manuscript technically sound, and do the data support the conclusions?

Reviewer #1: Yes

Reviewer #2: Yes

2. Has the statistical analysis been performed appropriately and rigorously?

Reviewer #1: Yes

Reviewer #2: No

3. Have the authors made all data underlying the findings in their manuscript fully available?

Reviewer #1: Yes

Reviewer #2: Yes

4. Is the manuscript presented in an intelligible fashion and written in standard English?

Reviewer #1: Yes

Reviewer #2: Yes

Reviewer #1: This is an impressive and very relevant piece of operational research that offers valuable real-world insight into the use of the BPaLM/BPaL regimen for drug-resistant tuberculosis (DR-TB) in Indonesia. The work is both timely and significant, particularly in light of the country’s heavy DR-TB burden and the World Health Organization’s recent endorsement of this shorter, all-oral regimen.

The manuscript is well written and logically organized, with clear objectives and a sound methodological approach. The study population is appropriately chosen, and the statistical analysis—especially the use of multivariate logistic regression—has been applied correctly to identify predictors of treatment success.

The results are compelling: the 88.8% treatment success rate, along with the identification of treatment adherence and nutritional improvement as major determinants, provides strong, actionable evidence for national TB programs. The discussion thoughtfully connects these findings with existing global and Indonesian data, offering realistic suggestions for strengthening patient support and adherence monitoring. Ethical oversight and data transparency are also well addressed.

In summary, this is a solid and well-executed study that adds real value to the growing evidence base on the BPaLM/BPaL regimen. It makes both a scientific and policy contribution and will be of great interest to clinicians, researchers, and program managers involved in DR-TB control.

Reviewer #2: The manuscript is well-structured and addresses a scientifically important topic. Nonetheless, a few areas would benefit from additional clarification to further strengthen the presentation.

1. The retrospective study design is appropriate for the research objectives; however, additional details regarding the inclusion and exclusion criteria would strengthen the methodological transparency. In particular, clarification is needed on how patients were allocated to the BPaL versus BPaLM regimen and whether specific baseline characteristics—such as linezolid intolerance, HIV co-infection, or other clinical factors—influenced regimen selection. Providing this information would enhance the reader’s understanding of potential selection bias and comparability between the treatment groups.

2. The statistical methods are described briefly. Please specify which tests were used for comparisons and justify their selection to improve clarity and reproducibility.

3. The manuscript reports adverse events but provides limited detail. Consider including a table summarizing the frequency and severity of key events (e.g., neuropathy, anemia, hepatotoxicity).

4. In the Discussion, please expand on operational challenges, such as drug supply, patient adherence, and monitoring, and discuss how these factors may impact the scale-up of the BPaL/BPaLM regimen within the national TB program.

**Do you want your identity to be public for this peer review?** For information about this choice, including consent withdrawal, please see our For information about this choice, including consent withdrawal, please see our Privacy Policy .

Reviewer #1: **Yes:** Dr. Kaushal Kumar DwivediDr. Kaushal Kumar Dwivedi

Reviewer #2: No

---

## [Author Response · Author response to Decision Letter 1]

18 Jan 2026

Response to Reviewers

Reviewer #1:

This is an impressive and very relevant piece of operational research that offers valuable real-world insight into the use of the BPaLM/BPaL regimen for drug-resistant tuberculosis (DR-TB) in Indonesia. The work is both timely and significant, particularly in light of the country’s heavy DR-TB burden and the World Health Organization’s recent endorsement of this shorter, all-oral regimen.

The manuscript is well written and logically organized, with clear objectives and a sound methodological approach. The study population is appropriately chosen, and the statistical analysis—especially the use of multivariate logistic regression—has been applied correctly to identify predictors of treatment success.

The results are compelling: the 88.8% treatment success rate, along with the identification of treatment adherence and nutritional improvement as major determinants, provides strong, actionable evidence for national TB programs. The discussion thoughtfully connects these findings with existing global and Indonesian data, offering realistic suggestions for strengthening patient support and adherence monitoring. Ethical oversight and data transparency are also well addressed.

In summary, this is a solid and well-executed study that adds real value to the growing evidence base on the BPaLM/BPaL regimen. It makes both a scientific and policy contribution and will be of great interest to clinicians, researchers, and program managers involved in DR-TB control.

We are pleased that the reviewer recognizes the timeliness and relevance of this operational research, particularly in the context of Indonesia's significant DR-TB burden and the recent implementation of BPaLM/BPaL regimens in our national program. We also appreciate the acknowledgment of our methodological rigor, the clinical significance of our findings, and the practical implications for TB program management. Thank you.

Reviewer #2:

The manuscript is well-structured and addresses a scientifically important topic. Nonetheless, a few areas would benefit from additional clarification to further strengthen the presentation.

1 The retrospective study design is appropriate for the research objectives; however, additional details regarding the inclusion and exclusion criteria would strengthen the methodological transparency. In particular, clarification is needed on how patients were allocated to the BPaL versus BPaLM regimen and whether specific baseline characteristics—such as linezolid intolerance, HIV co-infection, or other clinical factors—influenced regimen selection. Providing this information would enhance the reader’s understanding of potential selection bias and comparability between the treatment groups.

We appreciate this important comment regarding treatment allocation and potential selection bias. We have addressed this by adding two new sections, "Treatment Allocation and Regimen Selection" and "Inclusion and Exclusion Criteria," to the Methods. These sections now clearly describe the criteria for selecting BPaLM versus BPaL regimens in our study (Line 119-130 & Line 132-152). We hope that this addresses the reviewer's concern regarding clarity and reproducibility

Regarding the specific baseline characteristics mentioned:

Linezolid intolerance: No patients had documented linezolid intolerance at baseline, as all were linezolid-naïve

HIV co-infection: As described in the regimen selection criteria, the choice between BPaLM and BPaL regimens was not determined by HIV status. In our study, only 1 patient (0.6%) had HIV co-infection, reflecting the low HIV prevalence in our setting.

Other clinical factors: An important consideration in regimen selection was the exclusion of pregnant and breastfeeding women, as evidence for BPaLM/BPaL use in this population remains limited based on ongoing studies.

We acknowledge that the retrospective design and clinician-directed treatment allocation may introduce potential selection bias. We have added this limitation to the Discussion section and noted that comparative effectiveness studies with randomization would be needed to definitively address this concern. (Line 403-407).

2 The statistical methods are described briefly. Please specify which tests were used for comparisons and justify their selection to improve clarity and reproducibility

We have expanded the Statistical Analysis section to improve clarity and reproducibility. The revised section now specifies: independent t-test for normally distributed continuous variables and Mann-Whitney U test for non-normally distributed variables; chi-square test or Fisher's exact test (when expected cell counts <5) for categorical variables; and multivariate logistic regression with backward elimination for identifying independent predictors. Variables with p<0.25 in bivariate analysis were included in the multivariate model (Line 182-204).

3 The manuscript reports adverse events but provides limited detail. Consider including a table summarizing the frequency and severity of key events (e.g., neuropathy, anemia, hepatotoxicity).

Thank you for this constructive suggestion. We have included Table 4, which summarizes treatment-related side effects observed during BPaLM/BPaL treatment in our cohort (N = 132 patients with documented adverse events). The table presents the frequency of key adverse events, including gastrointestinal manifestations (93.1%), peripheral neuropathy (30.3%), headache/vertigo (15.9%), arthralgia (20.4%), skin reactions (10.6%), and hearing impairment (3.0%).

We acknowledge that our adverse event documentation focused on clinically significant events that were systematically monitored during monthly follow-up visits as part of our standard DR-TB program protocol. While we documented the occurrence and frequency of these adverse events, detailed severity grading for each event was not consistently recorded in the medical records, as clinical management decisions were individualized by our multidisciplinary team based on overall patient assessment rather than standardized severity scoring systems.

4 In the Discussion, please expand on operational challenges, such as drug supply, patient adherence, and monitoring, and discuss how these factors may impact the scale-up of the BPaL/BPaLM regimen within the national TB program

We appreciate this valuable suggestion to discuss operational challenges relevant to national program implementation. We have added a new section in the Discussion titled "Operational Challenges and Program Implementation" that addresses: drug supply, patient adherence support, monitoring, and scale-up implications (Line 327-355).

---

## [Decision Letter · Decision Letter 1]

1 Feb 2026

Dear Dr. Purwiga,

Thank you for submitting your manuscript to PLOS ONE. After careful consideration, we feel that it has merit but does not fully meet PLOS ONE’s publication criteria as it currently stands. Therefore, we invite you to submit a revised version of the manuscript that addresses the points raised during the review process.

**ACADEMIC EDITOR:** Please see the minor additional edits suggested by reviewer 2. These seem fair and reasonable.

We look forward to receiving your revised manuscript.

Kind regards,

Graeme Hoddinott, Ph.D

Academic Editor

PLOS One

Journal Requirements:

Reviewers' comments:

Reviewer's Responses to Questions

**Comments to the Author**

Reviewer #1: All comments have been addressed

Reviewer #2: All comments have been addressed

2. Is the manuscript technically sound, and do the data support the conclusions?

Reviewer #1: Yes

Reviewer #2: Yes

3. Has the statistical analysis been performed appropriately and rigorously?

Reviewer #1: No

Reviewer #2: Yes

4. Have the authors made all data underlying the findings in their manuscript fully available?

Reviewer #1: Yes

Reviewer #2: Yes

5. Is the manuscript presented in an intelligible fashion and written in standard English?

Reviewer #1: Yes

Reviewer #2: Yes

Reviewer #1: The authors have commendably incorporated the previous suggestions, resulting in a stronger manuscript. However, one significant methodological concern regarding statistical power remains, along with a minor typographical correction.

Specific Comments

1. Statistical Power and Model Stability:

The current analysis reports only 19 "unsuccessful" events. Given that the multivariate logistic regression model includes multiple covariates, there is a high risk of overfitting. Generally, a minimum of 10 events per predictor variable (EPV) is recommended to ensure model stability. This instability is reflected in the extremely wide Confidence Intervals observed in Table 2 (e.g., the adjusted Odds Ratio for adherence ranges from 1.44 to 28.5).

o Recommendation: Please address this limitation in the Discussion section. Furthermore, to improve the robustness of the results, consider employing penalized regression methods (such as Firth’s bias reduction) or simplifying the multivariate model to include only the one or two strongest predictors.

2. Typographical Error:

In Table 1 and the footnote of Table 2, the term "Body Mass Indeks" appears. Please correct the spelling of "Indeks" (Indonesian) to the standard English spelling, "Index."

Reviewer #2: The author has satisfactorily incorporated all the reviewers’ comments. The manuscript is now well written, clear, and coherent .

**Do you want your identity to be public for this peer review?** For information about this choice, including consent withdrawal, please see our For information about this choice, including consent withdrawal, please see our Privacy Policy .

Reviewer #1: **Yes:** Dr. Kaushal Kumar DwivediDr. Kaushal Kumar Dwivedi

Reviewer #2: No

---

## [Author Response · Author response to Decision Letter 2]

23 Feb 2026

Editor Comments Section

Please see the minor additional edits suggested by reviewer 2. These seem fair and reasonable.

We have carefully reviewed and addressed all comments from both reviewers. All suggested edits have been incorporated into the revised manuscript as detailed below. Thank you

Reviewers' comments:

Reviewer's Responses to Questions

Comments to the Author

1. If the authors have adequately addressed your comments raised in a previous round of review and you feel that this manuscript is now acceptable for publication, you may indicate that here to bypass the “Comments to the Author” section, enter your conflict of interest statement in the “Confidential to Editor” section, and submit your "Accept" recommendation.

Reviewer #1: All comments have been addressed

Reviewer #2: All comments have been addressed

2. Is the manuscript technically sound, and do the data support the conclusions?

Reviewer #1: Yes

Reviewer #2: Yes

3. Has the statistical analysis been performed appropriately and rigorously?

Reviewer #1: No

Reviewer #2: Yes

We have addressed this concern by adding explicit discussion of the statistical limitations (wide confidence intervals due to n=19 events) in the Discussion section, while justifying our multivariable modeling approach based on systematic review evidence. Thank you

4. Have the authors made all data underlying the findings in their manuscript fully available?

Reviewer #1: Yes

Reviewer #2: Yes

5. Is the manuscript presented in an intelligible fashion and written in standard English?

Reviewer #1: Yes

Reviewer #2: Yes

6. Review Comments to the Author

Please use the space provided to explain your answers to the questions above. You may also include additional comments for the author, including concerns about dual publication, research ethics, or publication ethics. (Please upload your review as an attachment if it exceeds 20,000 characters.

Reviewer #1: The authors have commendably incorporated the previous suggestions, resulting in a stronger manuscript. However, one significant methodological concern regarding statistical power remains, along with a minor typographical correction.

Specific Comments

1. Statistical Power and Model Stability:

The current analysis reports only 19 "unsuccessful" events. Given that the multivariate logistic regression model includes multiple covariates, there is a high risk of overfitting. Generally, a minimum of 10 events per predictor variable (EPV) is recommended to ensure model stability. This instability is reflected in the extremely wide Confidence Intervals observed in Table 2 (e.g., the adjusted Odds Ratio for adherence ranges from 1.44 to 28.5).

o Recommendation: Please address this limitation in the Discussion section. Furthermore, to improve the robustness of the results, consider employing penalized regression methods (such as Firth’s bias reduction) or simplifying the multivariate model to include only the one or two strongest predictors.

We sincerely thank the reviewer for this important methodological concern regarding statistical power and model stability. We have carefully considered this suggestion and respectfully provide our rationale for not implementing penalized regression methods in this analysis, while acknowledging the limitations posed by wide confidence intervals. This study has several important limitations. First, the relatively small number of unsuccessful treatment events (n=19) resulted in wide confidence intervals for the multivariate analysis (e.g., aOR for adherence: 1.44-28.5), reflecting limited statistical precision

Our multivariate model was constructed based on a comprehensive literature review identifying established risk factors for DR-TB treatment outcome, which is influenced by numerous interrelated factors across different domains (Johnson, 2009). From a clinical and programmatic perspective, restricting our model to only one or two predictors (as suggested for parsimony) would not adequately reflect the real-world complexity of DR-TB treatment.

Rather than artificially improving model precision through penalized methods, we believe in transparently reporting the uncertainty inherent in our data. The wide confidence intervals serve as an honest acknowledgment of the limited precision. Our findings are intended to inform programmatic implementation of BPaLM/BPaL regimens in Indonesia's national TB program. Program managers need to consider the full spectrum of factors that may influence treatment success.

To address the reviewer's concern, we have substantially expanded the limitations section to explicitly discuss the implications of wide confidence intervals and small event numbers (Line Number 350-361).

2. Typographical Error:

In Table 1 and the footnote of Table 2, the term "Body Mass Indeks" appears. Please correct the spelling of "Indeks" (Indonesian) to the standard English spelling, "Index."

Reviewer #2: The author has satisfactorily incorporated all the reviewers’ comments. The manuscript is now well written, clear, and coherent .

We sincerely thank the reviewer for identifying this typographical error. The term "Body Mass Indeks" has been corrected to "Body Mass Index" in both Table 1 and the footnote of Table 2 throughout the manuscript. Thank you for your detailed reviews.

---

## [Decision Letter · Decision Letter 2]

10 Mar 2026

Factors Predicting Successful Treatment Outcome with Novel BPaLM/BPaL Regimen in Individuals with Drug-Resistant Tuberculosis: Experience From Indonesia

PONE-D-25-51649R2

Dear Dr. Purwiga,

We’re pleased to inform you that your manuscript has been judged scientifically suitable for publication and will be formally accepted for publication once it meets all outstanding technical requirements.

Kind regards,

Graeme Hoddinott, Ph.D

Academic Editor

PLOS One

Additional Editor Comments (optional):

Reviewers' comments:

Reviewer's Responses to Questions

**Comments to the Author**

Reviewer #1: All comments have been addressed

Reviewer #2: All comments have been addressed

2. Is the manuscript technically sound, and do the data support the conclusions?

Reviewer #1: Yes

Reviewer #2: Yes

3. Has the statistical analysis been performed appropriately and rigorously?

Reviewer #1: Yes

Reviewer #2: Yes

4. Have the authors made all data underlying the findings in their manuscript fully available?

Reviewer #1: Yes

Reviewer #2: Yes

5. Is the manuscript presented in an intelligible fashion and written in standard English?

Reviewer #1: Yes

Reviewer #2: Yes

Reviewer #1: The authors have satisfactorily addressed all the concerns raised in the previous reviews. The methodological clarifications and other corrections have improved the clarity and rigor of the manuscript.

Reviewer #2: The authors have adequately addressed the comments raised during the previous round of review. The revisions have improved the clarity and quality of the manuscript.

**Do you want your identity to be public for this peer review?** For information about this choice, including consent withdrawal, please see our For information about this choice, including consent withdrawal, please see our Privacy Policy .

Reviewer #1: **Yes:** Dr. Kaushal Kumar DwivediDr. Kaushal Kumar Dwivedi

Reviewer #2: No

---

## [Editor Report · Acceptance letter]

PONE-D-25-51649R2

PLOS One

Dear Dr. Purwiga,

I'm pleased to inform you that your manuscript has been deemed suitable for publication in PLOS One. Congratulations! Your manuscript is now being handed over to our production team.

Kind regards,

on behalf of

Dr. Graeme Hoddinott

Academic Editor

PLOS One